# Association between serum zinc and serum neurofilament light chains: A population-based analysis

Jun Wei, Ye Xu, Yang Liu ◉ *

School of Basic Medical Sciences, Jilin Medical University, Jilin, China

* yl92@mail.ustc.edu.cn

## Abstract

### Background

While serum zinc levels are strongly associated with various neurological disorders, the relationship between serum zinc and axonal damage remains largely unexplored. Consequently, the objective of this study was to investigate whether an association exists between serum zinc levels and serum neurofilament light chain (sNfL) concentrations in the general population.

### Methods

Data from the National Health and Nutrition Examination Survey (NHANES) conducted during 2013–2014 were utilized for this study. We applied multiple linear regression and smoothed curve fitting methods to explore the connection between serum zinc levels and sNfL. Furthermore, subgroup analyses and interaction tests were performed to assess the consistency of this association among various populations.

### Results

This analysis included a total of 585 adults. After controlling for various confounding variables, we identified a significant negative association between serum zinc levels and sNfL (β = −0.62, 95% CI: −1.18, −0.05, p = 0.0321). Stratified analyses revealed that this negative association was significant only among individuals who were divorced, widowed, or separated, with no such association observed in other subgroups. This finding suggests that serum zinc levels may have a more pronounced impact on neurological health within these specific populations. Additionally, we identified an L-shaped association between serum zinc and sNfL, with an inflection point at 10.21 nmol/L.

**Data availability statement:** The data used in this study are publicly available from the National Health and Nutrition Examination Survey (NHANES) database: https://www.cdc.gov/nchs/nhanes/. The authors used data from the 2013–2014 survey cycle. All data are freely available and can be accessed without restriction.

**Funding:** The author(s) received no specific funding for this work.

**Competing interests:** The authors have declared that no competing interests exist.

## Conclusion

Our findings demonstrate an inverse association between serum zinc levels and sNfL concentrations among adults in the United States. This relationship is particularly pronounced in individuals who are divorced, widowed, or separated.

---

## 1. Introduction

Neurofilaments are specialized structures classified as type IV intermediate filament heteropolymers, unique to neurons, and composed of three distinct chains: light, medium, and heavy [1]. These proteins are synthesized within the neuronal cell body, known as the perikaryon, and are subsequently integrated into the axonal cytoskeleton, which is essential for maintaining the structural integrity of neurons [2]. In the human body, neurofilaments are primarily degraded through two cellular mechanisms: proteasomal degradation [3] and autophagic lysosomal pathways [4]. However, in instances of neuro-axonal injury, these neurofilaments can be released into the cerebrospinal fluid and subsequently into the bloodstream [5]. Among the various types of neurofilaments, the neurofilament light chain (NfL) has been identified as a crucial component that can spontaneously assemble to form the structural backbone of nerve fibers [6]. Its relatively small molecular weight and significant solubility render NfL a valuable potential biomarker for detecting neuro-axonal injury [7]. In recent years, NfL concentrations have gained widespread recognition as a highly sensitive diagnostic marker for various neurological disorders, such as multiple system atrophy, multiple sclerosis, Alzheimer's disease, and stroke [8–10]. Furthermore, NfL concentrations in the body not only serve as indicators for the diagnosis of these conditions but also hold promise for predicting the progression of neurological diseases and assessing the effectiveness of therapeutic interventions [11,12]. Despite this, there remains a paucity of research investigating the relationship between serum zinc levels and serum NfL (sNfL) concentrations in the general population.

Zinc is an essential micronutrient that plays a central role in the nervous system by modulating synaptic transmission and plasticity [13]. Disruption of zinc homeostasis, characterized by abnormal intracellular zinc ion concentrations, can impair neuronal signaling and has been associated with the underlying mechanisms of neurodegenerative conditions, such as Alzheimer's and Parkinson's diseases [14]. As both an antioxidant and anti-inflammatory trace element, zinc is crucial for neuroprotection during aging and disease progression [15]. Its regulatory effects on numerous enzymes further underscore its significance in supporting essential biochemical processes within the body [16]. Despite its recognized importance, zinc deficiency remains a major global health concern, with approximately 2 billion people worldwide failing to consume adequate zinc in their diets [17]. This concern is especially significant in older adults, who tend to be more susceptible to the negative consequences of a lack of zinc [18]. Studies show that insufficient zinc amounts can negatively affect neurological processes and elevate the likelihood of encountering neurodegenerative

disorders like Alzheimer's disease [19]. These findings underscore the urgent need to address zinc deficiency, particularly among vulnerable populations, to improve public health outcomes.

Additionally, it is important to consider zinc's interaction with other micronutrients such as copper. High levels of zinc consumption may disrupt the absorption of copper, which could result in copper deficiency myelopathy—an ailment marked by sensory ataxia and demyelination in the spinal cord [20,21]. This highlights the need for a balanced micronutrient intake when evaluating zinc's neuroprotective potential.

This research investigates the potential relationship between serum zinc levels and sNfL concentrations through a comprehensive survey conducted on adults in the United States. The objective is to provide a novel approach for the early prevention of neurodegenerative disorders and to explore their underlying mechanisms.

## 2. Materials and methods

### 2.1 Survey description

The National Health and Nutrition Examination Survey (NHANES) is a prominent research initiative that began in the early 1960s, aimed at assessing the health conditions and lifestyle behaviors of the American populace. Known for its innovative combination of surveys and physical health assessments, NHANES is one of the fundamental programs of the National Center for Health Statistics (NCHS), which functions under the auspices of the Centers for Disease Control and Prevention (CDC) and primarily focuses on delivering crucial health statistics for the country. All protocols for NHANES research underwent evaluation by the NCHS Research Ethics Review Board, with all subjects providing written informed consent before the study commenced. Because this research only involved publicly available de-identified data, there was no need for further ethical review. For more details about NHANES, please refer to the official website.

### 2.2 Study population samples

Data from the NHANES cycle of 2013–2014 was employed in this research, involving a total of 10,175 individuals. Upon the removal of samples lacking serum zinc data (n = 7,656) and those without sNfL data (n = 1,832), participants with incomplete information on covariates (n = 102) were also excluded. In the end, the study incorporated 585 participants aged 20–75 years. A comprehensive illustration of the study's screening methodology is shown in Fig 1.

### 2.3 Measurement of sNfL concentrations

In this research, consent for blood sample collection was obtained from 50% of the participants aged between 20 and 75 years. The examination of blood specimens employed acoustic emission technology via the Attelica immunoassay platform, which integrates acridinium chemiluminescence alongside paramagnetic particles to improve both the sensitivity and rapidity of the sNfL immunoassay method. Initially, the specimens are incubated with antibodies tagged with acridinium-ester (AE) that are designed to bind specifically to the NfL antigen. Following this, capture antibodies coated on paramagnetic particles (PMPs) are introduced, leading to the formation of complexes involving the antigens, AE-labeled antibodies, and PMPs. Any AE-labeled antibodies that remain unbound are subsequently removed and discarded. Afterward, acid and base solutions are introduced to initiate chemiluminescence, facilitating the quantification of light emissions. Throughout the analysis and measurement processes, there was a strict compliance with quality assurance protocols. In addition to the study samples, quality control specimens at low, medium, and high concentrations were evaluated every 8 hours, along with additional replicate samples, to verify the accuracy and dependability of the data obtained. The detection range for sNfL varies from 3.9 to 500 pg/mL.

### 2.4 Assessment of serum zinc levels

In this research, we analyzed serum sample data procured from the NHANES 2013–2014 cycle. The Centers for Disease Control and Prevention (CDC) National Environmental Health Sciences Laboratory measured the zinc concentrations in

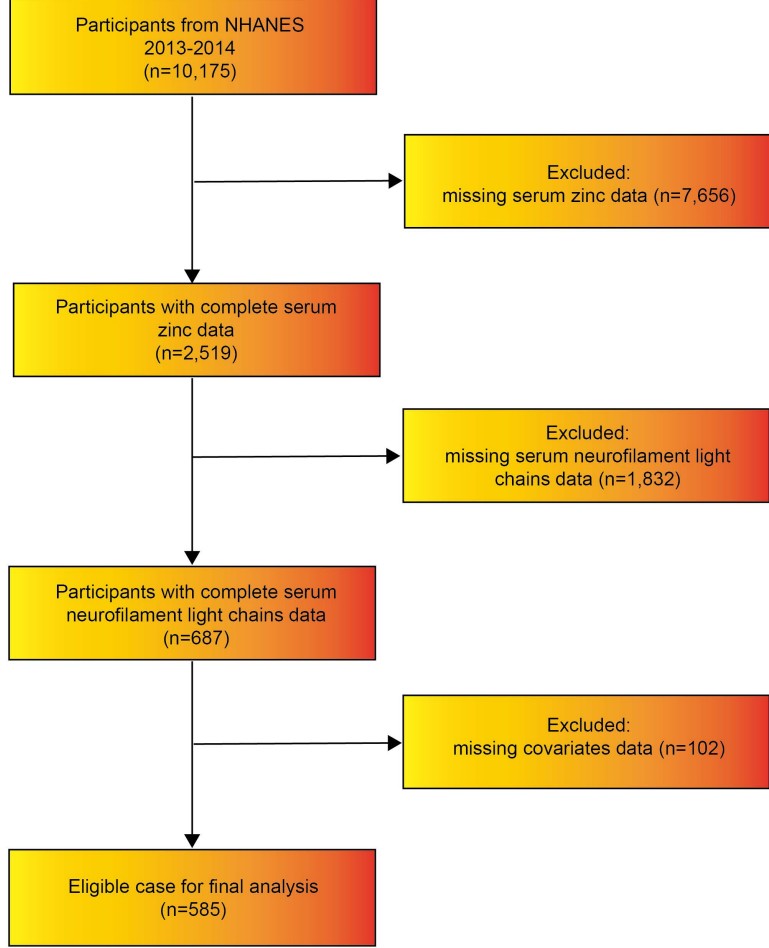

**Fig 1. Flowchart of participant screening and selection process.**

serum utilizing inductively coupled plasma dynamic reaction cell mass spectrometry (ICP-DRC-MS), which incorporates gallium as an internal standard and follows the NHANES protocol. A lower limit of detection (LOD) for serum zinc was determined to be 2.9 µg/dL, ensuring precise measurements. Rigorous protocols and guidelines were enforced for the collection, storage, and handling of samples to ensure the reliability of the data.

## 2.5 Covariates

The present study explores various demographic and health-related aspects, such as age (grouped as under 60 years and 60 years or older), gender, race/ethnicity (including non-Hispanic white, non-Hispanic black, and other categories), body mass index (BMI) categorized as normal weight (less than 25.0 kg/m²), overweight (ranging from 25.0 to 30.0 kg/m²), and obesity (30.0 kg/m² and above), levels of educational achievement (less than high school, high school graduates or their equivalents, and those with college education or higher), martial situation (married or cohabitating, never married, and divorced, widowed, or separated), and family poverty income ratio (PIR), categorized into <1.3, 1.3 to 3.5, and ≥3.5. Furthermore, factors such as smoking habits, the prevalence of diabetes, and alcohol intake were also assessed.

## 2.6 Statistical analysis

We employed linear regression models to evaluate the association between serum zinc and sNfL concentrations, adjusting for potential covariates. To effectively control for confounding variables, we constructed three regression models in this study: the original unadjusted model; Model 1, adjusted for age, sex, and ethnicity; and Model 2, which included further adjustments for education level, PIR, BMI, smoking, alcohol consumption, and diabetes. Additionally, We also divided serum zinc levels into quartiles to conduct sensitivity analyses and evaluate the reliability of the findings. Smoothed curve fitting was utilized to explore the nonlinear relationship between serum zinc and sNfL concentration. To analyze threshold effects, we compared the segmented regression model with the single-case model using a log-likelihood ratio test. Furthermore, to evaluate the stability of the association between serum zinc and sNfL concentration, we conducted subgroup analyses using stratified multifactorial regression analyses. All analyses were performed utilizing R (version 4.2) or EmpowerStats (version 5.0), with statistical significance defined as a P-value of 0.05.

## 3. Results

### 3.1 Baseline characteristics for all participants

This study included a total of 585 adults, with a mean age of 46.76 ± 15.54 years, comprising 289 males and 296 females. The study sample consisted of 49.4% males and 50.6% females. The characteristics of the participants are summarized by gender, as illustrated in the figure. The mean ages for males and females were 46.13 ± 15.49 years and 47.36 ± 15.60 years, respectively. Notably, the BMI was higher in the female group (29.91 ± 8.22) compared to the male group (28.13 ± 6.31). Furthermore, males were significantly more likely than females to consume alcohol and smoke cigarettes, with rates of 87.54% versus 64.53% and 54.33% versus 34.46%, respectively. Notable disparities were found between males and females in BMI, serum zinc levels, marital status, alcohol consumption, and smoking. Additionally, the differences between the male and female groups did not reach statistical significance for sNfL concentrations, age, PIR, race, education level, and diabetes status (all p > 0.05) (Table 1).

### 3.2 The association between serum zinc and sNfL levels

The findings from the multiple linear regression analyses are shown in Table 2. Serum zinc was found to have a negative association with sNfL in model 2 (β = −0.62, 95% CI: −1.18, −0.05, p = 0.0321). In contrast, both the unadjusted model and model 1 did not yield statistically significant results (p > 0.05). Specifically, for every 1 μmol/L increase in serum zinc, there was a corresponding decrease in sNfL of 0.62 pg/mL. When serum zinc was categorized into quartiles, individuals in the lowest quartile exhibited a decrease in sNfL of 3.25 pg/mL compared to those in the highest quartile (Table 2).

Smooth curve-fitting models revealed an L-shaped association between serum zinc levels and sNfL (Fig 2). Threshold effect analyses (Table 3) indicated that the lowest threshold for a protective effect was 10.21 (log-likelihood ratio < 0.001). Progressively decreasing odds of developing a neurological disorder were observed on the left side of the inflection point as serum zinc levels increased. (β = −12.19; 95% CI, −18.57, −5.81). In contrast, on the right side of the inflection point, there was neither a significant protective effect nor statistical significance (β = −0.24; 95% CI, −0.84, 0.35). Although not statistically significant, a visual inspection of Fig 2 reveals a slight upward trend in sNfL concentrations at higher serum zinc concentrations (beyond approximately 20 μmol/L).

### 3.3 Subgroup analysis

The association between serum zinc levels and sNfL varied significantly across different marital subgroups. A negative association was observed in the divorced, widowed, and separated group, whereas no such association was found in the other two groups. Notably, there were no significant differences in age, BMI, diabetes mellitus, alcohol consumption, smoking status, education level, PIR, or race across these subgroups (all p for interaction > 0.05) (Table 4).

**Table 1. Baseline characteristics of the participants.**

| Variables | Total (n = 585) | Male (n = 289) | Female (n = 296) | p-value |
|---|---|---|---|---|
| Age, years (mean (SD)) | 46.76±15.54 | 46.13±15.49 | 47.36±15.60 | 0.338 |
| BMI, kg/m2 | 29.03±7.39 | 28.13±6.31 | 29.91±8.22 | 0.004 |
| Serum Zinc (umol/L) | 13.50±2.33 | 13.83±2.33 | 13.17±2.28 | <0.001 |
| Race, n (%) | | | | 0.871 |
| Non-Hispanic white | 282 (48.21%) | 142 (49.13%) | 140 (47.30%) | |
| Non-Hispanic black | 93 (15.90%) | 44 (15.22%) | 49 (16.55%) | |
| Other Race | 210 (35.90%) | 103 (35.64%) | 107 (36.15%) | |
| Education, n (%) | | | | 0.054 |
| Less than high school | 118 (20.17%) | 59 (19.93%) | 59 (19.93%) | |
| High school or equivalent | 121 (20.68%) | 71 (24.57%) | 50 (16.89%) | |
| College or above | 346 (59.15%) | 159 (55.02%) | 187 (63.18%) | |
| Marital status | | | | <0.001 |
| Married/living with partner | 343 (58.63%) | 187 (64.71%) | 156 (52.70%) | |
| Never married | 124 (21.20%) | 63 (21.80%) | 61 (20.61%) | |
| Divorced/widowed/separated | 118 (20.17%) | 39 (13.49%) | 79 (26.69%) | |
| PIR (mean (SD)) | | | | 0.086 |
| <1.3 | 200 (34.19%) | 101 (34.95%) | 99 (33.45%) | |
| 1.3-3.5 | 196 (33.50%) | 85 (29.41%) | 111 (37.50%) | |
| ≥3.5 | 189 (32.31%) | 103 (35.64%) | 86 (29.05%) | |
| Smoke (%) | | | | <0.001 |
| Yes | 259 (44.27%) | 157 (54.33%) | 102 (34.46%) | |
| No | 326 (55.73%) | 132 (45.67%) | 194 (65.54%) | |
| Alcohol (%) | | | | <0.001 |
| Yes | 444 (75.90%) | 253 (87.54%) | 191 (64.53%) | |
| No | 141 (24.10%) | 36 (12.46%) | 105 (35.47%) | |
| Diabetes (%) | | | | 0.622 |
| Yes | 67 (11.45%) | 35 (12.11%) | 32 (10.81%) | |
| No | 518 (88.55%) | 254 (87.89%) | 264 (89.19%) | |
| sNFL (pg/ml) | 15.60±16.70 | 17.31±20.43 | 13.93±11.80 | 0.014 |

Mean±SD for continuous variables: the P value was calculated by the weighted linear regression model; (%) for categorical variables: the P value was calculated by the weighted chi-square test. Abbreviation: PIR, the ratio of income to poverty; BMI, body mass index; sNFL, serum neurofilament light chains.

**Table 2. Associations between serum zinc with sNfL.**

| Exposure | Non-adjusted model | Model 1 | Model 2 |
|---|---|---|---|
| Serum Zinc (umol/L) | β (95%CI) p | β (95%CI) p | β (95%CI) p |
| Continuous | −0.48 (−1.06, 0.10) 0.1064 | −0.49 (−1.03, 0.06) 0.0823 | −0.62 (−1.18, −0.05) 0.0321 |
| Q1 | Reference | Reference | Reference |
| Q2 | −0.97 (−4.80, 2.86) 0.6207 | −1.28 (−4.87, 2.31) 0.4846 | −1.70 (−5.29, 1.89) 0.3544 |
| Q3 | −3.32 (−7.17, 0.53) 0.0914 | −3.28 (−6.90, 0.34) 0.0761 | −3.54 (−7.18, 0.10) 0.0568 |
| Q4 | −1.96 (−5.80, 1.88) 0.3180 | −2.35 (−5.99, 1.28) 0.2049 | −3.25 (−6.95, 0.45) 0.0857 |
| *P* for trend | 0.0004 | 0.0006 | 0.0010 |

Model 1: no covariates were adjusted. Model 2: age, gender, and race were adjusted. Model 3: age, gender, race, education level, marital status, PIR, BMI, drinking alcohol, diabetes were adjusted. Abbreviation: PIR, the ratio of income to poverty; BMI, body mass index; sNFL, serum neurofilament light chains.

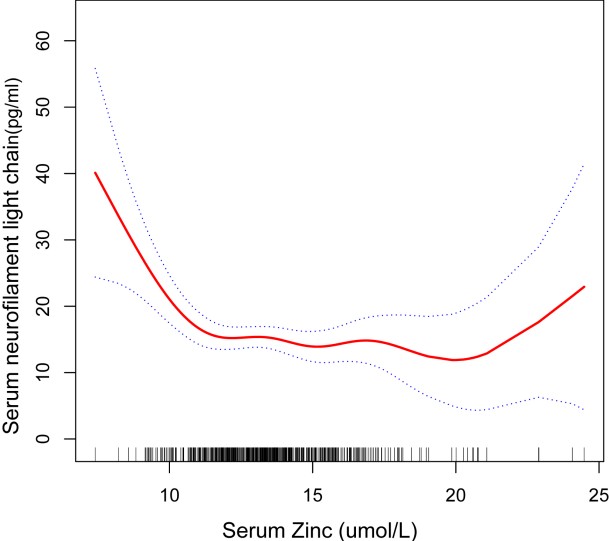

**Fig 2. The nonlinear associations between serum zinc with sNfL.** The solid red line represents the smooth curve fit between variables. Blue bands represent the 95% of confidence interval from the fit.

**Table 3. Analysis of threshold and saturation effects.**

| sNfL | β (95% CI) |
|---|---|
| | *P* value |
| Total | |
| Fitting by the standard linear model | −0.62 (−1.18, -0.05) 0.0321 |
| Fitting by the two-piecewise linear model | |
| Inflection point | 10.21 |
| Serum zinc<10.21(umol/L) | −12.19 (−18.57, −5.81) 0.0002 |
| Serum zinc>10.21(umol/L) | −0.24 (−0.84, 0.35) 0.4227 |
| Log likelihood ratio | <0.001 |

## 4. Discussion

This cross-sectional study enrolled 585 representative participants to investigate the relationship between serum zinc levels and sNfL. The findings reveal a significant negative association characterized by an L-shaped relationship, indicating a threshold-dependent effect of zinc. Specifically, below the threshold of 10.21 nmol/L, zinc appears to exert a protective effect on neuronal health, as evidenced by reduced sNfL concentrations.

As far as we are aware, this research is the initial investigation into the correlation between serum zinc concentrations and sNfL. Existing data suggests that lower serum zinc levels are closely linked to the development and advancement of several neurodegenerative disorders, especially Alzheimer's disease, Parkinson's disease, and multiple sclerosis. A particular study found a notable connection between a deficiency in serum zinc and cognitive deterioration [22]. Furthermore, a cohort study involving an elderly population found that serum zinc deficiency may be linked to the development of dementia in patients with Parkinson's disease [23]. Zinc deficiency is believed to exacerbate oxidative stress and inflammatory responses, leading to neurological damage that may further contribute to the progression of neurodegenerative diseases [24,25]. Our findings align with recent studies that emphasize the role of neurofilament light chains in aging-related neuronal damage [26] and the influence of trace element imbalances on neuroinflammation [27].

**Table 4. Subgroup analysis of the association between serum zinc with sNfL.**

| Participants | β (95% CI) | *P*-value | *P* for interaction |
|---|---|---|---|
| Age (categorical, years, n) | | | 0.7538 |
| <60 | −0.57 (−1.22, 0.09) | 0.0911 | |
| ≥60 | −0.76 (−1.82, 0.30) | 0.1595 | |
| Sex (n) | | | 0.3093 |
| Male | −0.86 (−1.64, −0.08) | 0.0309 | |
| Female | −0.30 (−1.10, 0.50) | 0.4613 | |
| BMI (kg/m²) | | | 0.5437 |
| <25 | −0.52 (−1.37, 0.34) | 0.2345 | |
| 25–30 | −0.39 (−1.47, 0.70) | 0.4828 | |
| ≥30 | −1.15 (−2.23, −0.06) | 0.0391 | |
| Diabetes (n) | | | 0.6915 |
| Yes | −1.01 (−2.64, 0.62) | 0.2246 | |
| No | −0.67 (−1.29, −0.05) | 0.0344 | |
| Alcohol use (n) | | | 0.6734 |
| Yes | −0.70 (−1.35, −0.06) | 0.0335 | |
| No | −0.43 (−1.59, 0.73) | 0.4678 | |
| Smoking status(n) | | | 0.0721 |
| Yes | −1.18 (−2.00, −0.36) | 0.0051 | |
| No | −0.19 (−0.93, 0.56) | 0.6227 | |
| Education (n) | | | 0.0586 |
| Less than high school | 0.14 (−0.99, 1.26) | 0.8130 | |
| High school or equivalent | −2.01 (−3.44, −0.57) | 0.0062 | |
| College or above | −0.63 (−1.39, 0.14) | 0.1105 | |
| PIR (n) | | | 0.3568 |
| <1.3 | −0.93 (−1.81, −0.05) | 0.0393 | |
| 1.3-3.5 | −0.91 (−2.10, 0.29) | 0.1367 | |
| ≥3.5 | −0.02 (−1.11, 1.07) | 0.9732 | |
| Race (n) | | | 0.3380 |
| Non-Hispanic white | −1.10 (−1.97, −0.24) | 0.0130 | |
| Non-Hispanic black | −0.34 (−1.71, 1.03) | 0.6228 | |
| Other Race | 0.24 (−1.17, 0.69) | 0.6136 | |
| Marriage (n) | | | 0.0400 |
| Married/living with partner | −0.41 (−1.16, 0.35) | 0.2896 | |
| Never married | 0.53 (−0.81, 1.88) | 0.4364 | |
| Divorced/widowed/separated | −1.65 (−2.82, −0.48) | 0.0058 | |

Age, gender, race, education level, PIR, BMI, drinking alcohol, diabetes were adjusted. Abbreviation: PIR, the ratio of income to poverty, BMI, body mass index.

Notably, although our threshold analysis did not yield statistically significant results above 10.21 μmol/L, a slight upward trend in sNfL concentrations was visually observed in Fig 2 at higher serum zinc concentrations. This suggests that elevated zinc levels may not be entirely benign. Previous research has indicated that excessive zinc intake can interfere with copper absorption, leading to copper deficiency and associated neurological complications, such as copper deficiency myelopathy. This condition has been described as clinically resembling subacute combined degeneration and is increasingly recognized as a consequence of chronic zinc excess [28]. Therefore, while zinc appears to have neuroprotective

properties, our findings highlight the importance of maintaining serum zinc levels within an optimal range to avoid potential toxicity.

Interestingly, our subgroup analyses revealed a pronounced relationship between serum zinc levels and sNfL, particularly among individuals who are divorced, widowed, or separated. A meta-analysis involving 800,000 participants found that individuals who remain lifelong unmarried face a 42% increased risk of developing dementia, while those who are widowed experience a 20% increased risk [29]. Furthermore, marital dissolution—such as divorce or widowhood—can result in diminished social support and heightened feelings of loneliness, both of which are associated with cognitive decline and an elevated risk of dementia [30].

In addition to psychosocial stressors, individuals experiencing marital dissolution may also face dietary inadequacies due to factors such as reduced meal regularity, lower dietary diversity, and limited nutritional support. These changes can lead to deficiencies in key micronutrients, including zinc and copper, which are essential for maintaining neuronal function. Previous population-based studies have reported that divorced, widowed, or separated individuals often have poorer nutritional status and higher rates of micronutrient deficiencies [31]. These nutritional vulnerabilities, combined with increased stress and social isolation, may partially explain the stronger association observed in this subgroup.

Our findings align with prior research regarding the neuroprotective potential of zinc in neurodegenerative disorders. Specifically, zinc deficiency has been associated with an increased risk and accelerated progression of Alzheimer's disease, likely due to its role in modulating amyloid beta (Aβ) aggregation and oxidative stress [32]. Conversely, elevated sNfL concentrations are widely accepted as indicators of axonal injury and disease progression across various neurological disorders [33]. Although our cross-sectional analysis cannot establish causality, the observed association between serum zinc levels and sNfL provides novel insights and suggests that zinc may function as a potential biomarker for neurodegenerative risk.

It is also important to consider that serum micronutrient concentrations, including zinc, reflect not only dietary intake but may also be influenced by broader health-related factors. These include poor nutrition, physical inactivity, chronic inflammation, comorbid conditions, and BMI. Therefore, zinc levels should not be interpreted solely as a measure of nutritional adequacy, but also as a proxy indicator of overall physiological status. This complexity has been similarly noted in studies of other micronutrients, such as vitamin D, where low levels may indicate poor general health beyond simple dietary deficiency [34].

The L-shaped association observed in our data parallels findings related to other nutrients. For instance, studies on vitamin D and calcium demonstrate that their protective associations also exhibit threshold effects, beyond which no additional benefits are observed. These patterns underscore the importance of identifying optimal, rather than maximal, nutrient levels in relation to neurological health [35].

In clinical practice, the measurement of sNfL concentrations may serve as an early and sensitive biomarker for neuronal injury, thereby facilitating timely intervention. Given the established association between serum zinc levels and sNfL, the exploration of zinc supplementation as a potential therapeutic strategy to mitigate neuronal damage is warranted, particularly for populations at risk of zinc deficiency. However, excessive zinc intake has been associated with adverse effects, including immune suppression and gastrointestinal disturbances [36,37]. Therefore, it is essential to establish evidence-based guidelines for optimal zinc intake.

Despite its strengths, including the utilization of a large, nationally representative dataset and robust statistical analyses, this study has several limitations. Initially, the cross-sectional approach prevents the establishment of causal relationships; therefore, conducting longitudinal research is crucial to verify the temporal link between zinc levels and sNfL. Moreover, this investigation depended on a singular assessment of serum zinc and sNfL, which might not fully reflect variations over time. Furthermore, the lack of dietary intake information limits the understanding of zinc sources and its adequacy. While a statistically significant negative association between serum zinc and sNfL was identified (p = 0.0321), this p-value should be regarded carefully due to the relatively small sample size. Although the significance observed is below

the standard threshold, it may lack robustness given the constraints of statistical power, necessitating future research involving larger cohorts to confirm these outcomes.

## 5. Conclusions

This study highlights a significant negative association between serum zinc levels and sNfL concentrations, indicating an L-shaped relationship that suggests a threshold-dependent neuroprotective effect. These findings emphasize the importance of maintaining adequate zinc levels for neuronal health and lay a foundation for future research focused on preventing and mitigating neurodegenerative diseases.

## Acknowledgments

**We would like to thank all participants in this study.**

## Author contributions

**Data curation:** Jun Wei.

**Investigation:** Ye Xu.

**Supervision:** Yang Liu.

**Validation:** Ye Xu.

**Writing – original draft:** Jun Wei.

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
