## [Decision Letter · Decision Letter 0]

PONE-D-25-01163

Association between serum zinc and serum neurofilament light chains: A population-based analysis

PLOS ONE

Dear Dr. Liu,

Thank you for submitting your manuscript to PLOS ONE. After careful consideration, we feel that it has merit but does not fully meet PLOS ONE’s publication criteria as it currently stands. Therefore, we invite you to submit a revised version of the manuscript that addresses the points raised during the review process.

We look forward to receiving your revised manuscript.

Kind regards,

Huzaifa Umar

Academic Editor

PLOS ONE

Journal Requirements:

Reviewers' comments:

Reviewer's Responses to Questions

**Comments to the Author**

1. Is the manuscript technically sound, and do the data support the conclusions?

Reviewer #1: Yes

Reviewer #2: Partly

2. Has the statistical analysis been performed appropriately and rigorously? 

Reviewer #1: Yes

Reviewer #2: Yes

3. Have the authors made all data underlying the findings in their manuscript fully available?

Reviewer #1: Yes

Reviewer #2: Yes

4. Is the manuscript presented in an intelligible fashion and written in standard English?

Reviewer #1: Yes

Reviewer #2: Yes

5. Review Comments to the Author

Reviewer #1: 1.technical and grammar are needed to be checked.

2. Percent of male and female was not presented in the study.

3. plagiarism report need to be submitted

4. two studies need to be acknowledged in this study; a- https://www.frontiersin.org/journals/aging-neuroscience/articles/10.3389/fnagi.2025.1517663/full b-https://pubmed.ncbi.nlm.nih.gov/32258830/

5. Graphs or tables need to be added

Reviewer #2: In this intriguing article, the authors investigated the association between serum zinc levels and serum neurofilament light chain (sNfL) levels in the general population, utilizing data from the National Health and Nutrition Examination Survey (NHANES) conducted in 2013-2014. Their analysis included a total population of 585 adults and revealed a significant negative association between serum zinc levels and sNfL, with a p-value at the margins of significance (0.0321). Stratified analyses indicated that this negative association was significant only among divorced, widowed, or separated individuals. Given these results, I believe the abstract's conclusion is somewhat overstated in asserting that serum zinc could serve as a novel predictor for the onset and progression of neurological disorders.

Zinc is an essential micronutrient that plays various roles in the nervous system, and both its deficiency and excess can lead to several issues. In my opinion, the introduction is overly generic and should delve deeper into the essential activities in which zinc is involved within the nervous system. I would eliminate the sections pertaining to wound healing, tissue reconstruction, growth, and immune system function, as they are not directly relevant and rather superficial. Furthermore, the interaction between zinc intake and other micronutrients (such as copper) is not addressed. The potential association of copper deficiency myelopathy, which may arise as a complication of excessive zinc supplementation, should be cited (see DOI: 10.1001/archneur.61.5.762 and DOI: 10.1179/2045772314Y.0000000268).

Other suggestions follow:

- Smooth curve-fitting models revealed an L-shaped association between serum zinc levels and sNfL, as depicted in Figure 2. Threshold effect analyses (Table 3) indicated that the lowest threshold for a protective effect was 10.21 nmol/L (log-likelihood ratio < 0.001). However, even if not statistically significant, a careful observation of Figure 2 reveals an increase on the right side of the curve, with higher serum concentrations of zinc associating with increased levels of serum NfL. I believe this is important since excessive zinc intake has been associated with decreased concentrations of copper.

- In the limitations section, it should be noted that for the size of the population, a significance level of p = 0.0321 is relatively low.

- The explanation of the subgroup analyses, which revealed that the relationship between serum zinc and sNfL was particularly pronounced among divorced, widowed, or separated individuals, appears insufficiently incisive. It is appropriate to highlight that changes in marital status can influence the onset and progression of neurodegenerative diseases, but the possible mechanisms should be better specified. Are there data on the diet and potential micronutrient deficiencies in this population of individuals?

- When discussing micronutrient and vitamin deficiencies and the incidence of disease, it should be acknowledged that low nutrient levels may reflect poor dietary habits, low levels of physical exercise, and can be influenced by many health factors, including body mass index. Therefore, micronutrient levels should not be considered solely as an indicator of nutritional status but also as a marker of overall health (e.g., for vitamin D, see DOI: 10.3390/nu7075111).

6. PLOS authors have the option to publish the peer review history of their article (what does this mean? ). If published, this will include your full peer review and any attached files.

**Do you want your identity to be public for this peer review?** For information about this choice, including consent withdrawal, please see our Privacy Policy .

Reviewer #1: **Yes: ** Umar Muhammad Ghali

Reviewer #2: **Yes: ** Domenico Plantone

---

## [Author Response · Author response to Decision Letter 1]

8 May 2025

Reviewer #1: 1.technical and grammar are needed to be checked.

Response: We appreciate the reviewer’s observation. The manuscript has been thoroughly revised for grammar, syntax, and technical clarity to ensure improved readability and language quality.

2. Percent of male and female was not presented in the study.

Response: Thank you for this suggestion. We have added the percentage distribution of male (49.4%) and female (50.6%) participants in the Results section for greater clarity.

3. plagiarism report need to be submitted

Response: We appreciate the opportunity to revise our manuscript. A plagiarism check was conducted using Turnitin, and the Similarity Index is 19%, which is within acceptable limits for originality. The plagiarism report has been submitted along with the revised manuscript for your reference.

4. two studies need to be acknowledged in this study; a- https://www.frontiersin.org/journals/aging-neuroscience/articles/10.3389/fnagi.2025.1517663/full b-https://pubmed.ncbi.nlm.nih.gov/32258830/

Response: We thank the reviewer for recommending these valuable references. Both studies have been carefully reviewed and are now appropriately cited in the second paragraph of the Discussion section to enhance the contextual background and support the mechanistic interpretation of our findings.

5. Graphs or tables need to be added

Response: Thank you for this helpful suggestion. We have ensured that key findings are now appropriately presented in visual form. Specifically:

A flowchart illustrating the participant selection process is provided (Figure 1).

A smooth curve fitting plot depicting the nonlinear relationship between serum zinc and sNfL is presented (Figure 2).

Detailed statistical outcomes are included in Tables 1–4. These additions enhance the readability and interpretability of the results.

Reviewer #2: In this intriguing article, the authors investigated the association between serum zinc levels and serum neurofilament light chain (sNfL) levels in the general population, utilizing data from the National Health and Nutrition Examination Survey (NHANES) conducted in 2013-2014. Their analysis included a total population of 585 adults and revealed a significant negative association between serum zinc levels and sNfL, with a p-value at the margins of significance (0.0321). Stratified analyses indicated that this negative association was significant only among divorced, widowed, or separated individuals. Given these results, I believe the abstract's conclusion is somewhat overstated in asserting that serum zinc could serve as a novel predictor for the onset and progression of neurological disorders.

Response: We thank the reviewer for this thoughtful and important comment. In response, we have revised the conclusion of the abstract to avoid overstatement. The updated sentence now reads:

Our findings demonstrate an inverse association between serum zinc levels and sNfL concentrations among adults in the United States. This relationship is particularly pronounced in individuals who are divorced, widowed, or separated.

We hope this revision better reflects the observational nature of our study and addresses the reviewer’s concern appropriately.

Zinc is an essential micronutrient that plays various roles in the nervous system, and both its deficiency and excess can lead to several issues. In my opinion, the introduction is overly generic and should delve deeper into the essential activities in which zinc is involved within the nervous system. I would eliminate the sections pertaining to wound healing, tissue reconstruction, growth, and immune system function, as they are not directly relevant and rather superficial. Furthermore, the interaction between zinc intake and other micronutrients (such as copper) is not addressed. The potential association of copper deficiency myelopathy, which may arise as a complication of excessive zinc supplementation, should be cited (see DOI: 10.1001/archneur.61.5.762 and DOI: 10.1179/2045772314Y.0000000268).

Response: We sincerely thank the reviewer for this insightful comment. In response:

We have revised the introduction to focus more specifically on the role of zinc in the nervous system, emphasizing its regulation of synaptic plasticity and neurotransmission.

General physiological roles unrelated to neurobiology—such as wound healing and immune function—have been removed to maintain thematic focus.

We have also incorporated a discussion on zinc’s interaction with copper and highlighted the risk of copper deficiency myelopathy resulting from excessive zinc intake. The recommended references (DOI: 10.1001/archneur.61.5.762 and DOI: 10.1179/2045772314Y.0000000268) have been cited appropriately.

We believe these revisions improve the scientific depth and focus of the manuscript.

Other suggestions follow:

Smooth curve-fitting models revealed an L-shaped association between serum zinc levels and sNfL, as depicted in Figure 2. Threshold effect analyses (Table 3) indicated that the lowest threshold for a protective effect was 10.21 nmol/L (log-likelihood ratio < 0.001). However, even if not statistically significant, a careful observation of Figure 2 reveals an increase on the right side of the curve, with higher serum concentrations of zinc associating with increased levels of serum NfL. I believe this is important since excessive zinc intake has been associated with decreased concentrations of copper.

Response: We thank the reviewer for this valuable observation. In response, we have revised the Discussion section to acknowledge the slight upward trend in sNfL levels at higher serum zinc concentrations, as visible on the right side of the curve in Figure 2, despite the lack of statistical significance. We discussed the potential implications of this trend and referenced existing literature on zinc-induced copper deficiency and its neurological consequences, including copper deficiency myelopathy. The following references have been cited accordingly:

Kumar N, Gross JB Jr, Kumar R. Copper deficiency myelopathy produces a clinical picture like subacute combined degeneration. Arch Neurol. 2004;61(5):762–765. doi:10.1001/archneur.61.5.762

These additions enhance the discussion by addressing possible risks of excessive zinc intake and the importance of maintaining optimal—not excessive—zinc levels for neurological health.

In the limitations section, it should be noted that for the size of the population, a significance level of p = 0.0321 is relatively low.

Response: We thank the reviewer for this thoughtful comment. In response, we have revised the Limitations section to note that the observed p-value (p = 0.0321), although statistically significant, may be relatively modest considering the sample size of the study. We now acknowledge that this finding should be interpreted with caution and highlight the need for larger-scale studies to confirm the observed association.

The explanation of the subgroup analyses, which revealed that the relationship between serum zinc and sNfL was particularly pronounced among divorced, widowed, or separated individuals, appears insufficiently incisive. It is appropriate to highlight that changes in marital status can influence the onset and progression of neurodegenerative diseases, but the possible mechanisms should be better specified. Are there data on the diet and potential micronutrient deficiencies in this population of individuals?

Response: We thank the reviewer for this thoughtful suggestion. In response, we have expanded the Discussion section to provide a more detailed explanation of the potential mechanisms underlying the pronounced association observed in divorced, widowed, or separated individuals. Specifically, we now discuss the possible roles of psychosocial stress, social isolation, and diet-related micronutrient deficiencies—factors that may contribute to increased vulnerability to neurodegenerative processes in this population. Existing literature supports the notion that individuals undergoing marital dissolution are at increased risk of poor diet quality and micronutrient deficiencies, which may further exacerbate neuronal decline.

When discussing micronutrient and vitamin deficiencies and the incidence of disease, it should be acknowledged that low nutrient levels may reflect poor dietary habits, low levels of physical exercise, and can be influenced by many health factors, including body mass index. Therefore, micronutrient levels should not be considered solely as an indicator of nutritional status but also as a marker of overall health (e.g., for vitamin D, see DOI: 10.3390/nu7075111).

Response: We thank the reviewer for this valuable and nuanced comment. In response, we have revised the Discussion section to acknowledge that micronutrient levels, including serum zinc, are influenced by a wide range of health-related factors beyond nutritional intake alone. Specifically, we now discuss how poor dietary habits, low physical activity, inflammation, comorbidities, and BMI can affect nutrient levels. As such, we emphasize that serum zinc should be interpreted not only as a marker of nutritional status, but also as a broader indicator of general health. We have also cited the suggested reference on vitamin D as a representative example (DOI: 10.3390/nu7075111).

---

## [Editor Report · Decision Letter 1]

Association between serum zinc and serum neurofilament light chains: A population-based analysis

PONE-D-25-01163R1

Dear Dr. Yang Liu,

We’re pleased to inform you that your manuscript has been judged scientifically suitable for publication and will be formally accepted for publication once it meets all outstanding technical requirements.

Kind regards,

Huzaifa Umar

Academic Editor

PLOS ONE
---

## [Editor Report · Acceptance letter]

PONE-D-25-01163R1

PLOS ONE

Dear Dr. Liu,

I'm pleased to inform you that your manuscript has been deemed suitable for publication in PLOS ONE. Congratulations! Your manuscript is now being handed over to our production team.

Kind regards,

on behalf of

Dr. Huzaifa Umar

Academic Editor

PLOS ONE